# Chemical Constituents and Anticancer Activities of Marine-Derived Fungus *Trichoderma lixii*

**DOI:** 10.3390/molecules29092048

**Published:** 2024-04-29

**Authors:** Natchanun Sirimangkalakitti, Jianyu Lin, Kazuo Harada, Andi Setiawan, Mitsuhiro Arisawa, Masayoshi Arai

**Affiliations:** 1Graduate School of Pharmaceutical Sciences, Osaka University, 1-6 Yamadaoka, Suita 565-0871, Osaka, Japan; siriman@phs.osaka-u.ac.jp (N.S.); lin-j@phs.osaka-u.ac.jp (J.L.); harada6@phs.osaka-u.ac.jp (K.H.); 2Department of Chemistry, Faculty of Science, Lampung University, Jl. Prof. Dr. Sumantri Brodjonegoro No. 1, Bandar Lampung 35145, Lampung, Indonesia; andi.setiawan@fmipa.unila.ac.id

**Keywords:** *Trichoderma lixii*, marine-derived fungus, trichodermamides, diketopiperazines, anticancer activity, antiproliferative activity, anti-austerity activity

## Abstract

The fungal genus *Trichoderma* is a rich source of structurally diverse secondary metabolites with remarkable pharmaceutical properties. The chemical constituents and anticancer activities of the marine-derived fungus *Trichoderma lixii* have never been investigated. In this study, a bioactivity-guided investigation led to the isolation of eleven compounds, including trichodermamide A (**1**), trichodermamide B (**2**), aspergillazine A (**3**), DC1149B (**4**), ergosterol peroxide (**5**), cerebrosides D/C (**6/7**), 5-hydroxy-2,3-dimethyl-7-methoxychromone (**8**), nafuredin A (**9**), and harzianumols E/F (**10**/**11**). Their structures were identified by using various spectroscopic techniques and compared to those in the literature. Notably, compounds **2** and **5**–**11** were reported for the first time from this species. Evaluation of the anticancer activities of all isolated compounds was carried out. Compounds **2**, **4**, and **9** were the most active antiproliferative compounds against three cancer cell lines (human myeloma KMS-11, colorectal HT-29, and pancreas PANC-1). Intriguingly, compound **4** exhibited anti-austerity activity with an IC_50_ of 22.43 μM against PANC-1 cancer cells under glucose starvation conditions, while compound **2** did not.

## 1. Introduction

Cancer remains a leading cause of mortality worldwide. From the latest global cancer data available in 2021, there were an estimated 19.3 million new cases and 10 million deaths [1], despite continuous efforts to better understand cancer biology and to improve diagnosis and treatment. The ongoing rise in cancer incidence and the limitations, failures, and excessive toxicity of conventional chemotherapies have generated the urgent need for the discovery and development of new anticancer agents [2]. 

Natural products derived from plants, animals, and microorganisms have a long history of usage in the treatment and prevention of human diseases, especially cancer and infectious diseases [2,3]. Remarkably, over half of all approved anti-cancer drugs between 1981 and 2019 were related to natural sources [4]. Among these, fungi stand as a significant source of secondary metabolites useful for drug discovery [5]. Numerous fungal metabolites and their analogs have advanced to various phases of cancer clinical trials [6]. 

The fungal genus *Trichoderma* comprises over 250 species [7] found in most types of soils, decaying wood, and in marine environments from various geographical regions and climatic zones [8,9]. *Trichoderma* spp. have promising industrial and agricultural potential and are most commonly used as potent biocontrol agents, such as biopesticides, biofertilizers, growth stimulants, and inducers of plant disease resistance [10,11]. In addition to their ecological effects, it is well known that *Trichoderma* are a prolific source of structurally diverse secondary metabolites, such as terpenes, polyketides, diketopiperazines, peptides, and alkaloids. These chemical entities possess a variety of biological properties, which include antimicrobial, anticancer, antimicroalgal, herbicidal, nematicidal, and enzyme inhibitory properties [8]. The unique structures and modes of action of *Trichoderma*’s secondary metabolites have inspired a great deal of research interest. It has been suggested that these compounds have the potential for development as new drug candidates [8]. Marine environments, with their extreme conditions and extraordinary ecosystems, have proven to be a treasure trove of biologically active compounds [12]. In the past few decades, the chemistry of marine-derived fungi has received increasing attention. Their potential to produce secondary metabolites that are not found in terrestrial microorganisms has become a subject of intense fascination. Approximately 80 compounds from the marine-derived fungi *Trichoderma* spp. Have been structurally described, and many of them have shown promising therapeutic properties in drug discovery [13]. 

*Trichoderma lixii*, which belongs to the *T. harzianum* species complex, was named and described in 2015 [14]. Very few studies regarding the chemistry and biological activity of *T. lixii* have been published so far [15,16,17]. Our interest was stimulated by the observation that solvent extracts exhibited not only significant antiproliferative activity against three cancer cell lines (human myeloma KMS-11, colorectal HT-29, and pancreas PANC-1) but also exhibited anti-austerity activity against PANC-1 cancer cells. This prompted us to investigate the chemical constituents and anticancer activities of *T. lixii* extract. 

In this study, we aim to delve deep into the chemical constituents of *T. lixii* isolated from marine sponges collected from Mentawai Island, Indonesia, as well as their potential anticancer activities. The scarcity of studies on the chemistry and biological activity of *T. lixii* underscores the significance of our research.

## 2. Results and Discussion

### 2.1. Bioassay-Guided Isolation, Structure Identification and Distribution in Nature

The marine-derived fungus *T. lixii* was cultivated and then extracted with acetone-based solvents. In a preliminary bioactivity study (Table 1), the acetone extract at 100 μg/mL induced moderate inhibition of cell proliferation among all KMS-11, HT-29, and PANC-1 cell lines (37.9–58.9% inhibition). Notably, it was selectively toxic to PANC-1 cells under glucose starvation conditions (79.51% cytotoxicity) compared to PANC-1 cells under normal glucose conditions (15.32% cytotoxicity). The active acetone extract was then partitioned between ethyl acetate (EtOAc) and water, yielding EtOAc and water-soluble extracts. The EtOAc extract at 100 μg/mL was able to significantly inhibit cell proliferation among all cancer cell lines (71.0–100.5% inhibition) and showed high potency and selectivity to PANC-1 cells under glucose starvation conditions (95.21% cytotoxicity), while the water-soluble extract was inactive. The bioactive EtOAc extract was subjected to *n*-hexane/methanol (MeOH) partitioning. The *n*-hexane layer was concentrated to obtain *n*-hexane extract. The MeOH layer was concentrated and further dissolved in EtOAc to obtain MeOH-1 extract. The residue that did not dissolve in EtOAc was dissolved in MeOH to obtain MeOH-2 extract. The MeOH-1 and MeOH-2 extracts exhibited higher potency in both assays compared to the *n*-hexane extract.

For reasons of sample quantity, potency, and selectivity (Table 1), we decided to continue the isolation of both bioactive MeOH-1 and MeOH-2 extracts. They were separately subjected to fractionation using various column chromatography techniques. The higher polarity MeOH-2 extract was isolated to obtain three bioactive sub-fractions (A through C). Notably, these sub-fractions showed high potency and selectivity to PANC-1 cells under glucose starvation conditions, particularly sub-fractions A and C. The isolation of sub-fraction A yielded four compounds including trichodermamide A (**1**), trichodermamide B (**2**) [18], aspergillazine A (**3**) [19,20,21], and DC1149B (**4**) [22]. Ergosterol peroxide (**5**) [23,24,25] and cerebrosides D/C (**6/7**) [26] were obtained from the isolation of sub-fractions B and C, respectively.

The lower polarity MeOH-1 extract was fractionated to afford bioactive sub-fraction D. Subsequently, it was further purified to yield 5-hydroxy-2,3-dimethyl-7-methoxychromone (**8**) [27], nafuredin A (**9**) [28,29], and harzianumols E/F (**10/11**) [30]. All isolated compounds were thoroughly characterized using a combination of spectroscopic techniques, including nuclear magnetic resonance (NMR) spectroscopy, high-resolution mass spectrometry (HR-MS), ultraviolet (UV) spectroscopy, and optical rotation measurements. Obtained spectroscopic data (Appendix A) were compared with data from the literature to confirm the structural assignments. Compounds **1**–**3**, **4**, **5**–**7**, and **8**–**11** were classified into modified dipeptides, epidithiodiketopiperazines, lipids and sterols, and polyketides, respectively (Figure 1). In the previous report, compounds **1**, **3**, and **4** were isolated from this fungal strain [16], while the isolation of the other compounds (**2** and **5**–**11**) was the first study of secondary metabolites of *T. lixii*.

Compound **1** has been found in several species of *Trichoderma* [16,18,31,32,33,34,35,36] and other genera of fungi [19,20,37,38]. The production of compound **2** correlated with NaCl in a seawater-based medium [18,19,33,39]. It is indicated that *Trichoderma* spp. are a valuable resource to explore trichodermamides. Compound **3**, possessing a sulfide linkage, has been discovered co-existing with compound **1** [16,19,20,34,39]. Compound **4** is a rare type of epithiodiketopiperazine, since the sulfide linkage connects the α- and β-positions of two amino acid residues. The Cl atom in the structure of compound **4** was incorporated from NaCl in a seawater-based medium [33]. To the best of our knowledge, compound **4** was specifically produced by cf. *T*. *brevicompactum* [22,33] and *T. lixii* [16].

Sterol **5** is a well-known, naturally occurring compound obtained from a variety of plants, marine organisms, and fungi, including some *Trichoderma* spp. [40,41]. Compounds **6**/**7** were a mixture of cerebrosides D (major) and C (minor) which differ from each other by a double bond at C-3′ and C-4′. Cerebrosides are a class of glycosphingolipids composed of a ceramide molecule, which includes a long-chain fatty acid linked to a sphingosine base, and a sugar molecule, typically glucose or galactose. The chemical structures of **6**/**7** were conclusively determined based on extensive analyses of NMR, high-resolution matrix-assisted laser desorption/ionization mass spectrometry (HR-MALDI-MS) and fragmentation patterns observed in liquid chromatography–mass spectrometry and tandem mass spectrometry (LC-MS/MS) spectra (Appendix A), as well as comparison with the reported data in the literature [26]. Cerebrosides can be found in diverse marine organisms and microorganisms [42]. Previously, **6**/**7** were isolated from *Trichoderma* spp. [43,44].

Chromone **8** was isolated from lichen mycobionts and fungi, including some *Trichoderma* spp. [45,46,47]. Omura and coworkers first reported compound **9** from the sponge-derived fungus *Aspergillus niger* in 2001 [28]. It is a specific NADH-fumarate reductase inhibitor, which is potentially a selective anthelmintic agent [48]. As its useful biological activity, a total synthesis of compound **9** has been established [49], and later its analogs were developed [50,51,52]. Compound **9** has been obtained from a variety of fungi, not only *Aspergillus* spp. [28,53] but also *Trichoderma* spp. [29,45,54,55]. Compounds **10**/**11** were enantiomers that could not be distinguished with NMR spectra but displayed a pair of peaks in chiral reversed-phase high-pressure liquid chromatography (RP-HPLC). Due to limitations in available resources, the purification of compounds **6/7** and **10**/**11** was halted and they were reported as a mixture. Compounds **10**/**11** are C_13_-polyketides first isolated from the sponge-derived fungus *T*. *harzianum*. The C_13_-polyketide metabolites are rarely found in natural sources and are a group of largely uninvestigated compounds embedded in fungi [30].

### 2.2. Antiproliferative Activity of Isolated Compounds against Three Cancer Cell Lines (KMS-11, HT-29, and PANC-1) and Human Umbilical Vein Endothelial Cells (HUVEC)

The assessment of antiproliferative effects is a fundamental in vitro assay in the early stages of drug discovery for the screening of potential anticancer compounds that merit further investigation and development for their therapeutic potential in the fight against cancer. The antiproliferative activity of all isolated compounds against three cancer cell lines (KMS-11, HT-29, and PANC-1) was examined by WST-8 based assay according to the manufacturer’s instructions.

As shown in Table 2, after incubation for 48 h, compounds **2**, **4**, and **9** showed strong antiproliferative activity toward all tested cancer cell lines. In particular, compound **2** showed promising activity, with IC_50_ values ranging from 0.7 to 3.6 μM against all cancer cells. These values are lower than those reported for cisplatin used as a positive control. Compound **2** is structurally closely related to compound **1**, which was completely inactive in all tested cell lines, indicating that the chlorine atom at C-5 is a crucial part of the pharmacophore. The chlorohydrin moiety at C-4 and C-5 may serve as a precursor to an epoxide, which could be the biologically active form of this molecule [18].

Notably, compound **5** has been studied for its potential anticancer properties, which might be attributed to multiple mechanisms such as apoptosis induction, arresting cell cycles, and suppressing cell migration [56,57]; however, different cancer cell lines behaved differently in response to compound **5** in our study (IC_50_ > 100 μM). A substantial number of cerebrosides have been reported in the literature so far [42]; however, a very small number have undergone more in-depth investigation for their potential against challenging diseases. It is interesting that compounds **6**/**7** selectively inhibited cell proliferation toward KMS-11 (IC_50_ 21.05 μg/mL).

Compound **8** showed weak or no antimicrobial and 2,2-diphenyl-1-picrylhydrazyl (DPPH) scavenging activities [46,58]. In this study, it showed no antiproliferative activity against all tested cancer cell lines (IC_50_ > 100 μM) as well. Compound **9** is known as an anthelmintic compound showing selective inhibition of complex I in helminth mitochondria [48]. In addition to anthelmintic activity, compound **9** exhibited antibacterial activity against *Escherichia coli*, as well as other antimicrobial properties which address ecological and environmental concerns [45,54,55,59]. Surprisingly, in previous studies compound **9** showed no cytotoxicity against several cancer cell lines, including mouse lymphoma L5178Y cell line at 28 μM [29], human cervical HeLa, colon HCT-116, breast MCF-7, lung A549, and leukemia K562 and HL-60 cancer cell lines at 50 μM [60]. On the other hand, in our study, compound **9** strongly inhibited cell proliferation against KMS-11, HT-29, and PANC-1 cancer cell lines with IC_50_ values ranging from 6.9 to 15.3 μM. Compound **9** is unstable in air because of the oxygen-labile conjugated diene units [61], causing unwanted chemical changes that may result in the loss of bioactivity. Understanding the oxygen sensitivity of compound **9** is important for improving chemical handling and storage to maintain its integrity and prevent degradation of the compound. In addition, our study is the first to demonstrate that rare C_13_-polyketides (**10**/**11**) inhibited cell proliferation of various cancer types (IC_50_ 16.4–39.3 μg/mL).

The antiproliferative activity of all eleven isolated compounds against normal cell line HUVEC was also examined by WST-8 based assay (Table 2). Regrettably, the assessment of active compounds revealed a significant challenge: while they demonstrated toxicity against cancer cells, they also exhibited undesirable effects on HUVEC. Further extensive structural modifications and in-depth structure–activity relationship studies are required to enhance efficacy and reduce the side effects on normal cells.

### 2.3. Anti-Austerity Activity of Isolated Compounds against PANC-1 Cancer Cell Line

Cancer cells alter their metabolism to promote growth, survival, proliferation, and long-term maintenance. Numerous microenvironmental factors, especially tumor nutrient levels, affect cancer cell metabolism [62]. Since cells in most solid tumors are subjected to strong microenvironmental stresses, including low nutrient and oxygen availability, such cancer cells must develop mechanisms to overcome these unfavorable growth conditions, and therefore eventually gain highly aggressive characteristics [63]. Pancreatic cancer is one of the most aggressive and highly lethal solid tissue malignancies, with a very low 5-year survival rate [64]. Unlike other types of cancer, pancreatic tumors are extremely hypovascular and have a supply of only a limited amount of essential nutrients and oxygen to sustain aggressive growth. Pancreatic cancer cells can develop tolerance to nutrient starvation and adapt to such microenvironmental niches. Such a mechanism of remarkable tolerance is known as “austerity”. There is evidence that tolerating nutrient insufficiency might be an important determinant of tumor progression [65]. The discovery of anti-austerity small molecules that can selectively target cancer cells in nutrient starvation conditions is a promising strategy for the treatment of pancreatic cancer. Anti-austerity agents are less toxic to nutrient-rich cancer cells and thus are expected to have fewer side effects than conventional anticancer drugs. PANC-1 cancer cells have extreme resistance to glucose starvation [65], making them a suitable model for studying an anti-austerity approach.

To evaluate anti-austerity activity, all isolated compounds (**1**–**11**) were tested for their cytotoxic effect against PANC-1 cells that were seeded at a high density under both DMEM no glucose (glucose −) and DMEM high glucose media (glucose +). We have searched for natural compounds that show cytotoxicity under glucose starvation conditions, and that are without cytotoxicity under normal glucose conditions. As shown in Table 3, after incubation for 12 h, compounds **2** and **4** showed higher cytotoxic activity against PANC-1 cells adapted to glucose starvation (IC_50_ 17.76 and 22.43 μM, respectively); however, compound **2** did not display selectivity under glucose starvation conditions, as evidenced by a similar IC_50_ value in normal glucose conditions. The selective index (SI) indicates the preferential cytotoxicity of a compound under glucose starvation conditions, with higher values representing greater selectivity. Compounds **3** and **9** demonstrated moderate cytotoxicity under glucose starvation conditions (IC_50_ 75.59 and 48.97 μM, respectively) and selectivity (SI ≥ 5 and SI = 2, respectively). On the other hand, compounds **1**, **5**–**8**, and **10**/**11**, as well as cisplatin, showed no activity against PANC-1 cells under both glucose starvation and normal glucose conditions (IC_50_ > 100 μM or μg/mL). These results indicated that compound **4**, with an SI of 10, emerged as a promising anti-austerity agent, selectively targeting PANC-1 cells adapted to glucose starvation. Our findings corroborate a previous report that identified compound **4**, isolated from *T. lixii*, as an anti-austerity compound against PANC-1 cells through the inhibition of endoplasmic reticulum (ER) stress signaling and complex II in the mitochondrial electron transport chain [16]. The observed differences in IC_50_ and SI values between the present study and the prior one [16] may be attributed to variations in the glucose starvation media used for the experiments. Our report supports the idea that *T. lixii* is a valuable resource for the production of an anti-austerity agent of remarkable potential and significance.

## 3. Materials and Methods

### 3.1. Instrumentation and General Experimental Techniques

^1^H- and ^13^C-NMR spectra were measured on a Varian Inova 600-II NMR spectrometer (Varian, Inc., CA, USA). Chemical shifts were referenced to the following residue solvent peaks: CDCl_3_ (δ_H_ 7.26/δ_C_ 77.1), DMSO-*d*_6_ (δ_H_ 2.50/δ_C_ 39.5), or CD_3_OD (δ_H_ 3.31/δ_C_ 49.0). HR-MALDI-MS were recorded on a SpiralTOF™ JEOL JMS-S3000 mass spectrometer (JEOL Ltd., Tokyo, Japan). Optical rotations were measured on a Jasco P-1020 polarimeter (JASCO Corporation, Tokyo, Japan) in CHCl_3_ or MeOH. Silica gel 60 F_254_ aluminum sheets (Merck, Germany) were used for thin layer chromatography (TLC), and spots on the TLC plate were detected under UV light (254 nm) and by spraying with *p*-anisaldehyde reagent. Silica gel 60 N, 40–50 μM (Kanto Chemical Co. Inc., Tokyo, Japan) was used for normal-phase flash column chromatography. Reversed-phase medium-pressure liquid chromatography (RP-MPLC) was conducted using a dual channel automated Smart Flash EPCLC-W-Prep 2XY system with a UV detector (Yamazen Corporation, Osaka, Japan). The medium-pressure ODS (C18) chromatographic column (Yamazen Corporation, universal column size L: 3.0 × 16.5 cm; M: 2.3 × 12.3 cm; S: 1.8 × 11.4 cm, pore size 120 Å, particle size 50 μm) was conditioned by first eluting with 100% MeOH or MeCN, then equilibrating with a suitable initial mobile phase. After dissolving the extract in the initial mobile phase, the solution was loaded in the ODS (C18) inject column (Yamazen Corporation, size M: 2.0 × 7.5 cm; S: 1.5 × 4.4 cm; SS: 1.3 × 3.1 cm) and separated by the gradient elution program. The fractions were collected automatically based on time. The purity of each collected fraction was determined by analytical RP-HPLC. An analytical RP-HPLC system was composed of an LC-20AD pump, a DGU-20A3R degasser, a CTO-20AC column oven, and an SPD-M20A diode array detector (Shimadzu Corporation, Kyoto, Japan). Separations were performed on an XBridge^®^ C18 (4.6 × 150 mm, 130 Å, 3.5 μm) coupled with an XBridge^®^ BEH C18 Vanguard^®^ Cartridge (3.9 × 5 mm, 130 Å, 3.5 μm, Waters Corporation, Milford, MA, USA). The mobile phase was composed of either MeOH or MeCN and H_2_O degassed by sonication. The flow rate was 0.5 mL/min at 25 °C, and the injections were carried out through a 20 μL-loop. Data analysis was performed by LabSolutions (Shimadzu Corporation).

### 3.2. Fungus Culture

The marine-derived fungus *T. lixii* strain 15G49-1 was isolated from an unidentified marine sponge which was collected from Mentawai Island, Indonesia, in 2015. The strain was identified as *T. lixii* by Techno Suruga Laboratory Co., Ltd. (Shizuoka, Japan) based on morphological characteristics and 5.8S ribosomal DNA sequencing. The *T. lixii* 15G49-1 was maintained on an MG agar plate [2% extract malt (Nacalai Tesque, Inc., Kyoto, Japan), 2% D-(+)-glucose (Nacalai Tesque, Inc.), 0.1% Bactopeptone (Becton Dickinson and Co., Sparks, MD, USA), 3.8% Marine Art SF-1 (Tomita Pharmaceutical, Tokushima, Japan), and 2% agar powder (Nacalai Tesque, Inc.)]. The fungal mycelium on the MG agar plate (1 cm^2^) was inoculated into 20 test tubes containing 10 mL each of MG broth (2% extract malt, 2% glucose, 0.1% Bactopeptone, and 3.8% Marine Art SF-1) and incubated at 30 °C with shaking at 180 rpm for 8 days. Each fungal inoculum was poured into a 500 mL Erlenmeyer flask containing 25 g of unpolished rice (Japan) and 50 mL of 3.8% Marine Art SF-1, with the addition of artificial sea salt. After that, they were statically cultivated in the solid rice medium for 14 days at 30 °C.

### 3.3. Extraction, Isolation and Structure Identification

After culturing under the conditions described above, the fungal extract was obtained by sonication for 30 min using acetone (180 mL each × 2) and mixed acetone: MeOH:EtOAc (4:2:1, 160 mL × 1), followed by combining and evaporating the organic solvents under reduced pressure to obtain a crude acetone extract. The extract was partitioned into a 200 mL EtOAc/200 mL water mixture (four times) to obtain EtOAc and water extracts. The bioactive EtOAc extract (13.7 g) was partitioned into a 200 mL *n*-hexane/200 mL MeOH mixture (four times) to obtain *n*-hexane (6.0 g) and MeOH extracts. The MeOH extract was separated into two extracts. One extract dissolved well in EtOAc (MeOH-1 extract, 5.2 g) and another extract did not dissolve in EtOAc but dissolved well in MeOH (MeOH-2 extract, 0.6 g).

Following the guidance of bioassays, the MeOH-2 extract was subjected to an RP-MPLC column eluted with 55% MeCN in H_2_O, 100% MeCN, and 100% EtOH to obtain three sub-fractions A (118.1 mg), B (41.6 mg), and C (93.3 mg), respectively. The most active sub-fraction A was subjected to RP-MPLC (24–38% MeOH in H_2_O) and then normal-phase (NP) silica gel (1–2% MeOH in CHCl_3_) columns to yield compounds **1** (1.0 mg), **2** (3.0 mg), **3** (9.6 mg), and **4** (11.0 mg). Purification of sub-fraction B using RP-MPLC (80–85% MeOH in H_2_O) followed by NP silica gel (30% EtOAc in *n*-hexane) columns gave **5** (0.9 mg), while purification of sub-fraction C using two NP silica gel columns (5–10% MeOH in EtOAc and then 8–10% MeOH in CH_2_Cl_2_) gave **6**/**7** (1.3 mg). The MeOH-1 extract was fractioned on an NP silica gel column eluted with 60–80% EtOAc in *n*-hexane to afford sub-fraction D (222.5 mg). Sub-fraction D was purified by using NP silica gel with 0–1% MeOH in CH_2_Cl_2_ and RP-MPLC columns with 40–70% MeOH in H_2_O to yield **8** (0.8 mg), **9** (5.1 mg), and **10**/**11** (2.0 mg). All isolated compounds were identified by analyses of NMR, HR-MS, LC-MS/MS, UV, and optical rotation values as well as spectroscopic comparison to data from the literature (Appendix A).

### 3.4. Anticancer Activities

#### 3.4.1. Cell Lines and Cell Culture

Human myeloma KMS-11, colorectal HT-29, and pancreatic PANC-1 cancer cells were obtained from the Japanese Collection of Research Bioresources (JCRB) Cell Bank (JCRB1179, Osaka, Japan), the American Type Culture Collection (ATCC, HTB-38, Manassas, VA, USA), and the Riken BioResource Research Center (BRC, RCB2095, Ibaraki, Japan), respectively. KMS-11, HT-29, and PANC-1 were cultured in Roswell Park Memorial Institute (RPMI)-1640 medium (Nacalai Tesque, Inc.), McCoy’s 5A medium (Gibco Thermo Fisher Scientific Inc., Waltham, MA, USA), and Dulbecco’s Modified Eagle Medium (DMEM) high glucose (Nacalai Tesque, Inc.), respectively. The culture media contained 10% fetal bovine serum (FBS, Equitech-Bio Inc., Kerrville, TX, USA) and 50 μg/mL kanamycin (Fujifilm Wako Pure Chemical Corporation, Osaka, Japan). To induce cell adaptation to nutrient starvation, PANC-1 was cultured in DMEM no glucose (Nacalai Tesque, Inc.) containing 10% dialyzed FBS (Equitech-Bio Inc.) and 50 μg/mL kanamycin. Human umbilical vein endothelial cells (HUVEC) were obtained from Kurabo Industries Ltd., Osaka, Japan (KE-4109) and cultured in HuMedia-EB2 (Kurabo Industries Ltd.) supplemented with HuMedia-EG (2% FBS, 10 ng/mL human epidermal growth factor, 1.34 μg/mL hydrocortisone, antibacterial agents, 50 μg/mL gentamicin and 50 ng/mL amphotericin B, 5 ng/mL human fibroblast growth factor-B, and 10 μg/mL heparin). Cells were incubated in a humidified incubator at 37 °C in an atmosphere of 5% CO_2_.

#### 3.4.2. Antiproliferative Assay

The antiproliferative activity against cancer and normal cell lines was investigated by a WST-8 based assay. Each cell line was plated in a 96-well flat bottom plate at a low density of 3000 cells/100 μL/well overnight and then treated with crude extracts (final concentration 100 μg/mL) or serially diluted compounds for 48 h. Compounds to be tested were dissolved in DMSO for stock solution (20 μg/mL) and the final concentration of DMSO was 0.5%. Then, 100 μL of corresponding medium containing 10% WST-8 (Nacalai Tesque, Inc.) was added and incubated for up to 4 h. The absorbance of the formazan products was measured at 450 nm using an iMark^TM^ microplate reader (Bio-Rad Laboratories, Inc. Hercules, CA, USA). Cell viability was obtained from three independent experiments. %Inhibition ± SE values of crude extracts and IC_50_ ± SE values of tested compounds were determined using Microsoft Excel 2019 (Microsoft Corporation, Redmond, WA, USA) and GraphPad Prism 9 (GraphPad Software, Boston, MA, USA). Cisplatin (Fujifilm Wako Pure Chemical Corporation) was used as a positive control.

#### 3.4.3. Anti-Austerity Assay

The anti-austerity activity against PANC-1 cell line was investigated by a WST-8 based assay according to the method described above. PANC-1 cells were plated in a 96-well flat bottom plate at a high density of 10,000 cells/100 μL/well in a DMEM high glucose medium overnight. The medium was removed, washed with phosphate-buffered saline (PBS, Nissui Pharmaceutical Co., Ltd., Tokyo, Japan) twice, and then replaced with either DMEM high glucose or DMEM no glucose. After incubation for 12 h, crude extracts (final concentration 100 μg/mL) or serially diluted compounds were added, and the cells were incubated for an additional 12 h. Cell viability was evaluated by incubation with 10% WST-8, and %cytotoxicity ± SE values of crude extracts and IC_50_ ± SE values of tested compounds were determined by the above method. Selective index (SI) was based on the difference between the IC_50_ values obtained from the high glucose and no glucose media. Antimycin A was used as a positive control.

## 4. Conclusions

A marine-derived fungus *T. lixii* has emerged as a valuable source of diverse chemical structures with significant potential in the field of cancer research. Through bioactivity-guided isolation, eleven compounds were isolated, and their structures were elucidated using a combination of spectroscopic techniques and comparison with data from the literature. Compounds **2** and **5**–**11** were reported for the first time from this species, adding to the novelty of our findings. We demonstrated that compound **4** triggered not only strong antiproliferative activity in different cancer cells but also cytotoxicity in PANC-1 cells under glucose starvation conditions selectively. Our results suggest that compound **4** might be a highly promising lead candidate for the development of new anticancer agents against various cancers, particularly for targeting pancreatic tumors based on an anti-austerity strategy. Overall, our study highlights the potential of secondary metabolites from *T. lixii* in cancer treatment and encourages further investigation of *T. lixii* as a source of bioactive molecules for drug discovery and development.

## Figures and Tables

**Figure 1 molecules-29-02048-f001:**
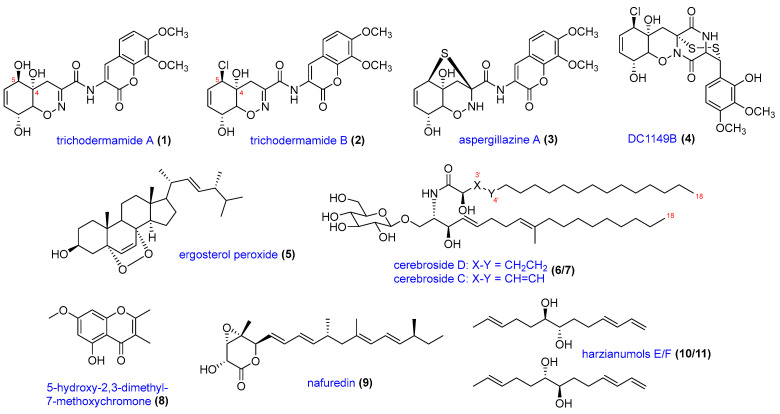
Eleven compounds isolated from the marine-derived fungus *Trichoderma lixii*.

**Table 1 molecules-29-02048-t001:** Anticancer activities of crude extracts and sub-fractions at 100 μg/mL.

Crude Extracts	Antiproliferative Activity (% Inhibition ± SE)			Anti-Austerity Activity (% Cytotoxicity ± SE)		
	KMS-11	HT-29	PANC-1	PANC-1		
				Glucose −	Glucose +	Selectivity
acetone	58.88 ± 4.37	53.11 ± 2.96	37.91 ± 4.07	79.51 ± 5.73	15.32 ± 0.68	selective
EtOAc	100.47 ± 1.00	90.44 ± 4.43	71.02 ± 4.78	95.21 ± 1.21	40.57 ± 6.16	selective
water	5.83 ± 5.85	3.96 ± 6.64	12.17 ± 5.43	−0.50 ± 4.63	0.92 ± 2.38	ND
*n*-hexane	72.42 ± 6.69	23.56 ± 6.09	30.44 ± 0.46	40.88 ± 4.20	22.12 ± 1.87	selective
MeOH-1	99.88 ± 1.30	93.48 ± 2.10	84.87 ± 2.83	100.53 ± 2.88	71.14 ± 2.80	selective
MeOH-2	75.93 ± 0.44	102.40 ± 1.59	99.54 ± 0.47	99.09 ± 0.43	65.71 ± 4.28	selective
sub-fraction A	NT			100.63 ± 2.03	41.26 ± 1.30	selective
sub-fraction B				62.99 ± 5.55	3.72 ± 6.38	selective
sub-fraction C				97.59 ± 2.21	26.13 ± 4.10	selective
sub-fraction D				68.75 ± 2.72	37.39 ± 5.66	selective

SE: standard error of mean of IC_50_; KMS-11: human myeloma cell line; HT-29: human colorectal cell line; PANC-1: human pancreas cell line; NT: not tested; Glucose −: under glucose starvation conditions (DMEM no glucose media); Glucose +: under normal glucose conditions (DMEM high glucose media); ND: not determined (no activity under Glucose −; thus, selectivity cannot be determined).

**Table 2 molecules-29-02048-t002:** Antiproliferative activity of isolated compounds (**1–11**) against three cancer cell lines (KMS-11, HT-29, and PANC-1) and HUVEC.

Compounds	Antiproliferative Activity (IC_50_ ± SE)							
	KMS-11		HT-29		PANC-1		HUVEC	
**1**	>100	μM	>100	μM	>100	μM	>100	μM
**2**	1.38 ± 0.12	μM	0.73 ± 0.12	μM	3.58 ± 0.32	μM	1.69 ± 0.26	μM
**3**	37.97 ± 7.72	μM	22.20 ± 3.09	μM	50.29 ± 4.21	μM	26.20 ± 5.51	μM
**4**	7.60 ± 0.40	μM	39.18 ± 4.60	μM	24.67 ± 4.44	μM	6.94 ± 0.86	μM
**5**	>100	μM	>100	μM	>100	μM	>100	μM
**6/7**	21.05 ± 3.21	μg/mL	>100	μg/mL	>100	μg/mL	31.27 ± 2.99	μg/mL
**8**	>100	μM	>100	μM	>100	μM	>100	μM
**9**	6.90 ± 1.84	μM	11.88 ± 0.55	μM	15.27 ± 0.44	μM	2.45 ± 0.16	μM
**10/11**	16.41 ± 2.23	μg/mL	32.51 ± 1.68	μg/mL	39.27 ± 2.57	μg/mL	15.32 ± 0.64	μg/mL
Cisplatin	4.91 ± 1.06	μM	10.59 ± 0.83	μM	28.62 ± 3.43	μM	13.72 ± 2.57	μM

IC_50_: 50% inhibitory concentration; SE: standard error of mean of IC_50_; KMS-11: human myeloma cell line; HT-29: human colorectal cell line; PANC-1: human pancreas cell line; HUVEC: human umbilical vein endothelial cells.

**Table 3 molecules-29-02048-t003:** Anti-austerity activity of isolated compounds (**1**–**11**) against PANC-1 cancer cells.

Compounds	Anti-Austerity Activity (IC_50_ ± SE)				
	Glucose −		Glucose +		Selectivity
**1**	>100	μM	>100	μM	ND
**2**	17.76 ± 4.75	μM	11.58 ± 1.59	μM	not selective
**3**	75.59 ± 4.08	μM	>400	μM	selective (SI ≥ 5)
**4**	22.43 ± 2.50	μM	213.37 ± 4.07	μM	selective (SI = 10)
**5**	>100	μM	>100	μM	ND
**6/7**	>100	μg/mL	>100	μg/mL	ND
**8**	>100	μM	>100	μM	ND
**9**	48.97 ± 1.31	μM	76.66 ± 3.22	μM	selective (SI = 2)
**10/11**	>100	μg/mL	>100	μg/mL	ND
Antimycin A	16.23 ± 0.26	nM	>300	μM	selective (SI ≥ 18,480)
Cisplatin	>100	μM	>100	μM	ND

IC_50_: 50% inhibitory concentration; SE: standard error of mean of IC_50_; Glucose −: under glucose starvation conditions (DMEM no glucose media); Glucose +: under normal glucose conditions (DMEM high glucose media); SI: selective index; ND: not determined (no activity under Glucose −, thus, selectivity cannot be determined).

## Data Availability

Data are contained within the article and Appendix A.

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
