# Peer review of "Chemical Constituents and Anticancer Activities of Marine-Derived Fungus Trichoderma lixii"

_molecules, 2024, doi:10.3390/molecules29092048_

Round 1
Reviewer 1 Report
Comments and Suggestions for Authors
The current contribution by Sirimangkalakitti et al., was focused on the chemical constituents and anticancer properties of the marine fungus, T. lixii. Activity-guided fractionation was used to isolate 11 notable compounds with anti-proliferative potential. Their structures were subsequently elucidated by various spectroscopic techniques. The authors of this work are worthy of commendation for an excellent and well-thought-out research with clear objective. The methods were appropriate and properly described. Importantly, the results and discussion are fact-based and presented with clarity, modesty and sufficient depth. In all, the presentation is excellent and there is no doubt that the findings described in this work holds promise for anti-cancer drug discovery based on various modifications of compounds 2, 4 and 9. Prior to publication, the authors are encouraged to consider the following points below.
1. All instances of the phrase ‘T. lixii’ in the manuscript should be italicized.
-Source of the cells used should be included (Section 3.4.1).
2. The % Similarity is quite high and unacceptable. Authors should check the manuscript again for plagiarism and ensure that sections are written in a manner that improves the originality of the text
3. Importantly, the results revealed that most potent anti-proliferative compounds identified were also cytotoxic against HUVEC (normal) cells. Given that only one normal cell line was used in this regard, the toxicity of the compounds on normal cells is a bit doubtful. This could be clarified by including perhaps two other normal cell lines in this work.
Good luck!
Author Response
1. All instances of the phrase ‘T. lixii’ in the manuscript should be italicized.
Answer: We have revised all instances of "T. lixii" to ensure they are appropriately italicized throughout the manuscript.
This modification has been made on
- [Page 2, Section 2.1, Paragraph 1]
- [Page 6, Section 2.3, Paragraph 2]
- [Page 7, Section 3.2]
-Source of the cells used should be included (Section 3.4.1).
Answer: We included the source of the cells used in Section 3.4.1 [Page 8].
2. The % Similarity is quite high and unacceptable. Authors should check the manuscript again for plagiarism and ensure that sections are written in a manner that improves the originality of the text
Answer: We have checked the similarity using iThenticate and found a 55% similarity report. Upon closer examination, we determined that this similarity primarily due to the "reference" section and "Instrumentation and General Experimental Techniques", which were adapted from our previous publication and involved the use of same specific instruments and reagents. However, considering the main context, we can confidently confirm the originality and lack of plagiarism in our work.
- Importantly, the results revealed that most potent antiproliferative compounds identified were also cytotoxic against HUVEC (normal) cells. Given that only one normal cell line was used in this regard, the toxicity of the compounds on normal cells is a bit doubtful. This could be clarified by including perhaps two other normal cell lines in this work.
Answer: Thank you for bringing up this important point. We recognize the significance of evaluating cytotoxicity against multiple normal cells. However, due to the very limited amounts of compounds we possess, conducting investigations in other normal cells is a very challenging situation. While our result of normal cells is limited to just one cell line, we believe we provide very valuable insights in terms of comparing cancer cells with normal cells.

Reviewer 2 Report
Comments and Suggestions for Authors
The manuscript detailing the isolation and characterization of secondary metabolites from the marine-derived fungus Trichoderma lixii presents a noteworthy exploration into the pharmaceutical potential of fungal compounds. The discovery of eleven compounds, particularly trichodermamide B, ergosterol peroxide, and nafuredin A, which exhibited significant antiproliferative activity against various cancer cell lines, is commendable. The anti-austerity activity of DC1149B under glucose starvation conditions is an intriguing finding that could have implications for cancer treatment strategies. However, the lack of novel structures among these compounds diminishes the manuscript’s impact. For the Journal that emphasizes novel discoveries, this could be a significant drawback. In addition, the manuscript acknowledges the existence of previous reports on some of the compounds. It is essential to clearly articulate how the current study adds value beyond what is already known. So, in its current form, the manuscript may not meet the novelty criteria for 'Molecules'. However, with thorough revisions that emphasize the unique aspects and potential applications of the findings, it could become suitable for publication.
Comments on the Quality of English Languageno
Author Response
Answer: Thank you for your comment. Considering only the structures of compounds, indeed they are all known compounds. However, for several compounds, we have, for the first time revealed that they are produced by this fungus. This finding can be considered highly significant in the context of natural product chemistry research focused on the producing organism. Furthermore, this study demonstrates novelty in comprehensively isolating secondary metabolites of Trichoderma lixii and elucidating their structures. In addition to these, discovering compounds with strong or unusual activity is extremely important, but it is also crucial to elucidate which compounds possess or lack specific activities. In this regard, we are providing new insights into the bioactivity profiles of these compounds.
We have emphasized the unique aspects and potential applications of the findings in conclusions [page 9]
Reviewer 3 Report
Comments and Suggestions for Authors
The experimental article “Chemical constituents and anticancer activities of marine-derived fungus Trichoderma lixii” is devoted to studying the potential of marine fungi Trichoderma lixii as producers of antitumor compounds. The topic of this research is extremely relevant due to the increase in mortality due to cancer. The article is written in high-quality academic language, logical and well structured, but there are few shortcomings:
1. The abstract states that “Evaluation of all isolated compounds revealed significant antiproliferative activity against three cancer cell lines”, however, such activity was not shown for compounds 1, 5, 6/7, 8. In this regard, this proposal must be corrected, since it is not supported by experimental data.
2. In section 2.1., it is recommended to transfer information regarding the separation of extracts and fungal cultivation conditions to the Materials and Methods section. In addition, it is not clear how antiproliferative and anti-austerity activities were determined for crude extracts, since the methods do not describe these experiments.
3. It is not clear what “-” means in tables 1 and 3, please explain in the footnotes.
4. Title of subchapter 3.1. “General Experimental Procedures” does not correspond to the contents of the chapter. It is recommended to rename.
5. It is not clear why in the list of references under numbers there are divisions into a, b, c. Please bring the in-text links and References into compliance with the requirements of the journal.
6. Please align the columns in Table 2.
7. In the text, “T. lixii” is often not written in italics (for example, pp. 2, 5, 6). Please correct it to italics.
Author Response
1. The abstract states that “Evaluation of all isolated compounds revealed significant antiproliferative activity against three cancer cell lines”, however, such activity was not shown for compounds 1, 5, 6/7, 8. In this regard, this proposal must be corrected, since it is not supported by experimental data.
Answer: Thank you for your feedback. We acknowledge the discrepancy in our abstract regarding the antiproliferative activity of certain isolated compounds. The statement "Evaluation of all isolated compounds revealed significant antiproliferative activity against three cancer cell lines" has been revised. We have made the necessary corrections [Page 1, Abstract] to ensure the accuracy of our findings.
2. In section 2.1., it is recommended to transfer information regarding the separation of extracts and fungal cultivation conditions to the Materials and Methods section.
Answer: We’ve transferred the information regarding the separation of extracts and fungal cultivation conditions [Pages 2-3, Section 2.1 Bioassay-guided isolation and structure identification, Paragraphs 1-3] to the Materials and Methods section.
In addition, it is not clear how antiproliferative and antiausterity activities were determined for crude extracts, since the methods do not describe these experiments.
Answer: We’ve provided descriptions of the methods used to determine antiproliferative and antiausterity activities for crude extracts [Pages 8-9, Sections 3.4.2. Antiproliferative assay and 3.4.3. Anti-austerity assay].
3. It is not clear what “-” means in tables 1 and 3, please explain in the footnotes.
Answer: We have changed from "-" to "ND" and included explanatory footnotes in Table 1 [Page 3] and Table 3 [Page 7] to clarify the meaning of "ND" to ensure better understanding for readers.
ND: not determined (no activity under Glucose -, thus, selectivity cannot be determined).
4. Title of subchapter 3.1. “General Experimental Procedures” does not correspond to the contents of the chapter. It is recommended to rename.
Answer: We’ve reviewed the contents of Section 3.1 and modified the title from "General Experimental Procedures" to "Instrumentation and General Experimental Techniques" to accurately reflect its contents [Page 7].
5. It is not clear why in the list of references under numbers there are divisions into a, b, c. Please bring the in-text links and References into compliance with the requirements of the journal.
Answer: We have reviewed and corrected the formatting of references to align with the journal's requirements [Pages 9-14]. Additionally, we removed divisions such as "a, b, c" in the reference list.
6. Please align the columns in Table 2.
Answer: We’ve aligned the columns in Table 2 and Tale 3 [Pages 5-7] for better presentation and readability.
7. In the text, “T. lixii” is often not written in italics
Answer: We have revised all instances of "T. lixii" to ensure they are appropriately italicized throughout the manuscript.
This modification has been made on
- [Page 2, Section 2.1, Paragraph 1]
- [Page 6, Section 2.3, Paragraph 2]
- [Page 7, Section 3.2]
Round 2
Reviewer 2 Report
Comments and Suggestions for Authors
Before the manuscript can be further consideration as an article in Molecules, there are several points that need to be addressed to ensure the completeness and clarity of the data presented:
1. NMR Data Presentation: The NMR data for all compounds should be presented in a tabular format in the Supplementary Information (SI). This will facilitate a quicker reference and comparison among the compounds studied.
2. Inclusion of NMR Spectra: It is essential to include the 1H and 13C NMR spectra for all compounds in the SI. These spectra are fundamental to the structural analysis and should be readily accessible to readers for verification.
3. HRESIMS Data: The high-resolution electrospray ionization mass spectrometry (HRESIMS) data for all compounds must be included in the SI. This data is crucial for confirming the exact mass of the compounds and thus their molecular formulae.
4. Structure Elucidation in Main Text: The section on structure elucidation found on page 4 requires reorganization. The current arrangement may lead to confusion, and a more logical sequence of presenting the data would greatly improve the understanding of the structural analysis performed.
5. Bioassay Results Precision: Bioassay results, especially for studies involving natural products, should be presented with two significant figures, along with the standard errors. The presentation of IC50 values as whole numbers lacks the precision expected in scientific reporting and raises questions about the accuracy of the measurements.
Author Response
Responses to reviewer 2’s comments:
Before the manuscript can be further consideration as an article in Molecules, there are several points that need to be addressed to ensure the completeness and clarity of the data presented:
- NMR Data Presentation: The NMR data for all compounds should be presented in a tabular format in the Supplementary Information (SI). This will facilitate a quicker reference and comparison among the compounds studied.
Answer: The NMR data for each isolated compound has been compiled and organized into a table (Table S1-S9) within the Supplementary Information (SI).
- Inclusion of NMR Spectra: It is essential to include the H and C NMR spectra for all compounds in the SI. These spectra are fundamental to the structural analysis and should be readily accessible to readers for verification.
Answer: We included detailed proton and carbon NMR spectra for each isolated compound within the SI.
- HRESIMS Data: The high-resolution electrospray ionization mass spectrometry (HRESIMS) data for all compounds must be included in the SI. This data is crucial for confirming the exact mass of the compounds and thus their molecular formulae.
Answer: We included HRMS data (HR-MALDI-MS or HR-ESI-MS) for each isolated compound within the SI.
- Structure Elucidation in Main Text: The section on structure elucidation found on page 4 requires reorganization. The current arrangement may lead to confusion, and a more logical sequence of presenting the data would greatly improve the understanding of the structural analysis performed.
Answer: We have carefully reorganized structure elucidation section to enhance understanding and readability.
- topic has been changed from "Bioassay-guided isolation and structure identification" to "Bioassay-guided isolation, structure identification, and distribution in nature"
- We incorporated additional details regarding the analytical methods employed for compound identification on Page 3, Paragraph 2
"All isolated compounds were thoroughly characterized using a combination of spectroscopic techniques, including nuclear magnetic resonance (NMR), high-resolution mass spectrometry (HR-MS), Ultraviolet (UV) spectroscopy, and optical rotation measurements. Obtained spectroscopic data (Supplementary Information) were compared with literature data to confirm the structural assignments"
- Bioassay Results Precision: Bioassay results, especially for studies involving natural products, should be presented with two significant figures, along with the standard errors. The presentation of IC50 values as whole numbers lacks the precision expected in scientific reporting and raises questions about the accuracy of the measurements.
Answer: We have revised the presentation of IC50 values in Tables 1-3 on Pages 2-3 and 5-7, as well as in the relatedtext. The updated tables now include IC50 values, along with their standard errors (SE), with two significant figures.

Round 3
Reviewer 2 Report
Comments and Suggestions for Authors
The paper is acceptable in its current form and requires no further corrections